# Expression of the Heterotrimeric GP2/GP3/GP4 Spike of an Arterivirus in Mammalian Cells

**DOI:** 10.3390/v14040749

**Published:** 2022-04-01

**Authors:** Anna Karolina Matczuk, Minze Zhang, Michael Veit, Maciej Ugorski

**Affiliations:** 1Division of Microbiology, Department of Pathology, Faculty of Veterinary Medicine, Wrocław University of Environmental and Life Sciences, 50-375 Wrocław, Poland; 2Institute of Virology, Department of Veterinary Medicine, Free University Berlin, 14163 Berlin, Germany; minze.zhang@fu-berlin.de (M.Z.); michael.veit@fu-berlin.de (M.V.); 3Department of Biochemistry and Molecular Biology, Faculty of Veterinary Medicine, Wrocław University of Environmental and Life Sciences, 50-375 Wrocław, Poland; maciej.ugorski@upwr.edu.pl

**Keywords:** equine arteritis virus, EAV, multi-cassete expression, mammalian expression, GP2/GP3/GP4, minor proteins, Arterivirus

## Abstract

Equine arteritis virus (EAV), an enveloped positive-strand RNA virus, is an important pathogen of horses and the prototype member of the Arteiviridae family. Unlike many other enveloped viruses, which possess homotrimeric spikes, the spike responsible for cellular tropism in Arteriviruses is a heterotrimer composed of 3 glycoproteins: GP2, GP3, and GP4. Together with the hydrophobic protein E they are the minor components of virus particles. We describe the expression of all 3 minor glycoproteins, each equipped with a different tag, from a multi-cassette system in mammalian BHK-21 cells. Coprecipitation studies suggest that a rather small faction of GP2, GP3, and GP4 form dimeric or trimeric complexes. GP2, GP3, and GP4 co-localize with each other and also, albeit weaker, with the E-protein. The co-localization of GP3-HA and GP2-myc was tested with markers for ER, ERGIC, and cis-Golgi. The co-localization of GP3-HA was the same regardless of whether it was expressed alone or as a complex, whereas the transport of GP2-myc to cis-Golgi was higher when this protein was expressed as a complex. The glycosylation pattern was also independent of whether the proteins were expressed alone or together. The recombinant spike might be a tool for basic research but might also be used as a subunit vaccine for horses.

## 1. Introduction

Equine arteritis virus (EAV) is the prototype Arterivirus, a virus family of veterinary importance [1]. EAV infects horses and donkeys and leads to abortions in pregnant mares and respiratory illness with flu-like symptoms, which can even lead to death in young animals. The virus is transmitted via the respiratory route or via contaminated semen of persistently infected stallions. Despite available vaccines, EAV remains an important pathogen in the horse industry [2].

EAV is an enveloped positive-stranded RNA virus. The structural proteins of EAV include the nucleocapsid protein N and seven membrane proteins [1,3]. The most abundant envelope proteins are GP5 and M, which form a disulphide-linked dimer [4]. GP5/M together with N are responsible for budding and virus particle formation [5]. Unlike in many other enveloped viruses, where the spike is a homotrimer, the spike responsible for cellular tropism in Arteriviruses is a heterotrimer composed of 3 different glycoproteins: GP2, GP3, and GP4 [6,7].

The other envelope components are small proteins: the myristoylated E protein, which might be an ion channel; and the product of the ORF5a gene, a protein of unknown function [8,9,10]. All structural proteins, except ORF5a, are essential for EAV replication in cell culture [5,9]. GP2 and GP4 are type I integral membrane proteins, which from a disulphide-linked dimer inside cells [3]. GP3 is a peculiar protein with an uncleaved signal peptide, which does not form a membrane anchor. Membrane anchoring is achieved by a C-terminal hydrophobic domain, which does not span the lipid bilayer but peripherally attaches GP3 to membranes [11,12]. In addition, GP3 is disulphide-linked to either Gp2 or Gp4, but disulphide-bond formation does not occur inside cells, only extracellularly in budded virus particles [13]. Despite its small size, GP3 is heavily glycosylated with 5 to 6 carbohydrates, whereas GP4 contains 2 and Gp2 1 N-linked carbohydrate (GP2 is 227 aa, GP3 is 163 aa, and GP4 is 152 aa for the reference Bucyrus strain) [3,11].

No 3D structure of any of the envelope proteins of any arterivirus has been resolved [3]. Basic knowledge on the functional domains of arteriviral spike, such as the locations of the receptor-binding domain and the fusion peptide, are also missing.

For porcine respiratory and reproductive syndrome virus (PRRSV), a virus from the same family, it was shown that a recombinant PRRSV, containing E and GP2/GP3/GP4 from EAV, exhibits the broadened cellular tropism of EAV [7]. In PRRSV, the GP2/GP3/GP4 complex was shown to attach to the CD163 cellular receptor on porcine alveolar macrophages [14].

The receptor for EAV on equine cells was identified as CXCL16, but it was not experimentally shown whether GP2/GP3/GP4 or GP5/M bind to the receptor [15]. It is also unknown what receptors are utilized by EAV in cell culture. In comparison to other arteriviruses, EAV has a wider tropism for primary cells and cell lines, such as Vero, RK-13, and BHK-21 [1].

After infection with EAV, the minor proteins are expressed in high amounts inside cells, where they co-localize at internal membranes but are not transported to the plasma membrane [13,16]. The precise co-localization with individual organelles of the secretory pathway has been investigated only for GP3 and E in virus-infected cells [17]. Despite the high expression levels in infected cells, only small amounts of GP2/GP3/GP4 are incorporated into virions [6]. The mechanism of minor protein retention in the secretory pathway and how their incorporation into virus particles is regulated remain unknown. It is unknown if the expression of the whole GP2/GP3/GP4 complex changes the intracellular localization and trafficking of the viral proteins. For other viral spike proteins, it has been well described that oligomerization is required for trafficking to the site of virus assembly [18].

The aim of this study was to express arterivirus spike components GP2/GP3/GP4 in mammalian cells. Here, we present the construction, cloning, and validation of the expression of the spike.

## 2. Materials and Methods

### 2.1. Cells and Media

The cell line BHK-21 (baby hamster kidney cells; ATCC C13) was maintained as an adherent culture in DMEM mixed in a 1:1 ratio with Leibovitz L-15 medium (Cytogen, Lodz, Poland), which was supplemented with 5% fetal calf serum (FCS) (Biological Industries, Cromwell, USA), 100 U of penicillin per ml, and 100 mg of streptomycin and L-glutamine per ml (Biowest, Lodz, Poland). The cells were maintained at 37 °C in an atmosphere of air with 5% CO_2_ and 95% humidity.

### 2.2. Construction of Plasmids, Cloning

The tagged sequences of EAV glycoproteins were created by overlap extension PCR from the reverse genetics plasmid pEAV211, which is a derivate of pEAV030 (Genbank Y07862.2) [19].

GP2 was tagged with myc tag (EQKLISEEDL), GP3 with HA tag (YPYDVPDYA), and GP4 with FLAG tag (DYKDDDK), which were fused directly to the C-terminus of the protein. In 1 construct, GP4 was separated from the FLAG tag with the linker GMAPGRDPPVAT. Upon unsuccessful tagging with FLAG, the V5 tag (GKPIPNPLLGLDST) was added to GP4 and it was also separated with the above-mentioned linker. The linker sequence was chosen from experiments on GP4-YFP and derives from vector pEYFP-N1-GP4 [11]. The E protein remained untagged.

Primers used for cloning are listed in Appendix A. The PCR products were cloned into MultiMam (Geneva Biotech, Geneva, Switzerland) or MultiLabel pML-DAZ2 from ATG:Biosynthetics (Merzhausen, Germany). Specifically, GP4-FLAG, GP4-linker-FLAG, and GP4-V5 were cloned into the acceptor vector pACEMam2 with XhoI and KpnI; GP2-myc into the donor vector pMDC with XbaI and BamHI and into the donor vector pMDK with KpnI and XhoI; and GP3-HA into the donor vector pMDS with KpnI and XhoI. The E gene was cloned without a tag into pML-DAZ2 with MluI and XbaI and into pMDK with KpnI and XhoI. pML-DAZ2 is a 3577-bp-long plasmid with CMV promotor and ampicillin resistance; its sequence is available from the authors upon request. The acceptor and donor vectors (GP2-myc in pMDK; GP3-HA in pMDS and GP4-linker V5 in pACEMam2) were combined with Cre-lox recombination as described by the manufacturer (New England BioLabs, Frankfurt am Main, Germany), and clones selected upon antibiotic selection: kanamycin, spectinomycin, and gentamycin (Sigma-Aldrich, Poznań, Poland). The donor plasmids were amplified in *E. coli* pir+ strain (Thermo Fisher Scientific, Warsaw, Poland). The acceptor and Cre-combined plasmids were amplified in *E. coli* DH5α. Plasmid DNA was purified (Extractme Plasmid Midi Endotoxin Free, Blirt, Gdańsk, Poland), control digested to check the presence of inserts, and fragments covering cloned genes were sequenced (Genomed, Warsaw, Poland) before use in experiments. The multicassete plasmid containing 3 genes of the EAV spike was named pGP2/GP3/GP4. A scheme of the successful cloning strategy is depicted in Figure 1.

### 2.3. Transient Expression in Mammalian Cells

Subconfluent BHK-21 cells were transfected with 2.5 μg of plasmid per 6-well dish with Lipofectamine 2000 (Thermo Fisher Scientific, Warsaw, Poland). For co-transfection of E cloned into pMDK with pGP2/GP3/GP4, 1.5 μg of each plasmid were used.

### 2.4. SDS-PAGE and Western Blotting

First, 22 h post transfection, the detached and low-speed centrifuged, transfected, and mock-transfected cells were directly boiled in 2× SDS-PAGE loading buffer (Bio-Rad, Warsaw, Poland) with DTT (Sigma-Aldrich, Poznań, Poland) and subjected to SDS-PAGE using 15% polyacrylamide. Then, the gels were blotted onto polyvinylidene difluoride (PVDF) membrane (GE Healthcare, Warsaw, Poland) using a Trans-Blot Turbo device (Bio-Rad, Warsaw, Poland). After blocking of the membranes (blocking solution; 5% skim milk powder in PBS with 0.1% Tween 20 (PBST)) overnight at 4 °C, the antibodies in the blocking solution were incubated for 1.5 h at room temperature. To detect the tags attached to each viral protein, the following antibodies were used: rabbit anti-HA tag antibodies (1:6000); ab9110; Abcam, Cambridge, UK) were used to detect GP3 with the HA tag, mouse monoclonal anti-V5 antibody (1:4000, ab27671, Abcam, Cambridge, UK), rabbit anti-myc (ab9106 Abcam, Cambridge, UK), rabbit anti-GFP (1:1000, D5.1, Cell Signaling Technologies, USA), rabbit anti-β-actin (13E5, Cell Signaling Technologies, Danvers, MA, USA), and rabbit anti-E (1:1000, described in [16], a gift from Eric Snijder, University of Leiden, Belgium).

After washing (3 times for 10 min each with PBST), suitable horseradish peroxidase-coupled secondary antibodies (1:8000; anti-rabbit or anti-mouse; Cell Signaling Technology, Danvers, MA, USA) were applied for 1 h at room temperature. After washing with PBST, the signals were detected by chemiluminescence using the ECL plus reagent (Thermo Fisher Scientific, Warsaw, Poland), and visualized in ChemiDoc (Bio-Rad, Warsaw, Poland).

### 2.5. Glycosidase Treatment

Transfected and mock-transfected cells were washed with PBS, detached from the dish with trypsin–EDTA (Biological Industries, Warsaw, Poland), pelleted, washed with PBS and resuspended in 50 µL of 1× glycoprotein denaturing buffer, and boiled for 10 min at 100 °C. Typically, 15 µL of this lysate were digested with Peptide-N-Glycosidase (PNGase F, 2.5–5 units/µL) or endoglycosidase H (Endo H, 2.5–5 units/µL) according to the manufacturer’s instructions (New England BioLabs, Ipswich, MA, USA) for 1 h at 37 °C. After the deglycosylation reaction, samples were supplemented with reducing SDS-PAGE buffer and subjected to SDS-PAGE and Western blot.

### 2.6. Immunoprecipitation

The BHK-21 cells seeded in the 6-well plate were transfected with GP2/GP3/GP4 or mock transfected. Then, 22 h p.t., cells were scraped, and resuspended either in Pierce-IP Lysis Buffer (25 mM Tris-HCl pH 7.4, 150 mM NaCl, 1 mM EDTA, 1% NP-40, and 5% glycerol) (Thermo Fisher Scientific, Warsaw, Poland), which is modified RIPA without SDS, or in the same formulated buffer with 1% DDM (n-dodecyl-β-D-maltopyranoside) (Sigma-Aldrich, Merck, Poland), as a detergent. Lysis buffers were supplemented with a complete protease inhibitor tablet (Roche, Merck, Poland).

Cells were lysed with agitation for 30 min at 4 °C and later centrifuged at 16,000*g* for 20 min at 4 °C. The supernatants were mixed with 1 µL of antibodies: rabbit anti-HA tag antibodies (ab9110; Abcam, Cambridge, UK), mouse monoclonal anti-V5 antibody (ab27671, Abcam, Cambridge, UK), and rabbit anti-myc antibody (ab9106 Abcam, Cambridge, UK), and shaken overnight at 4 °C. The antibody-protein complexes of the rabbit antibodies were pulled down with protein-A-Sepharose (Sepharose A) (Sigma-Aldrich, Poznań, Poland) while anti-V5 complexes were pulled down with protein-G-Sepharose (Sepharose G) (Sigma-Aldrich, Poznań, Poland), for 2.5 h. The beads were washed with the corresponding IP buffers 4 times, boiled with reducing SDS buffer, and subjected to SDS-PAGE and Western blotting as described above.

### 2.7. Immunofluorescence Assay

BHK-21 cells were seeded in complete medium on glass coverslips in 24-well plates. After 24 h, the cell culture medium was replaced with Opti-Mem (Lonza) and cells were transfected with 0.5 µg of the indicated plasmid or mock transfected using Lipofectamine 2000. For the experiments, cells were infected with EAV, and infection was carried out first. Subconfluent cell monolayers on coverslips were infected with the Bucyrus strain of EAV at MOI 1. After 2 h, the medium was removed, cells washed 2 times with PBS containing magnesium and calcium, and transfection with pGP2/Gp3/GP4 was performed as described above. At 22 h post-transfection, coverslips were subjected to immunofluorescence as described in [17].

Fixed, permeabilized, and blocked cells were incubated with the primary antibodies: a rabbit polyclonal anti-HA tag antibodies (1:500, ab9110, Abcam, UK); mouse monoclonal anti-HA tag antibody (Enzo Life Sciences, Warsaw, Poland); mouse monoclonal anti-V5 tag (ab27671, Abcam, Cambridge, UK); rabbit anti-myc antibody (ab9106, Abcam, Cambridge, UK); and mouse anti-myc antibody (ab18185, Abcam, Cambridge, UK), diluted in a blocking solution at room temperature for 1 h. These cells were then washed 3 times with PBS and incubated with secondary antibodies (1:800 goat anti-mouse IgG H&L Alexa Fluor 568, and 1:800 goat anti-rabbit IgG H&L Alexa Fluor 488, Abcam, UK). After each antibody incubation, cells were washed three times with PBS. Finally, the stained cultures were mounted on glass slides in a fluoroshield mounting medium with DAPI (Abcam, Cambridge, UK) and stored at 4 °C.

The confocal images were recorded using a Nikon inverted microscope eclipse Ti with fliters: SDC-405 for DAPI, excitation 405 nm, emission 460/50 nm; SDC-488 for green, excitation 488 nm, emission 525/59 nm; and SDC-561 for red, excitation 561 and emission 609/54 nm. Subsequently, the images were deconvoluted with Huygens Essential software (Scientific Volume Imaging, Hilversum, The Netherlands).

### 2.8. Co-Localization Assay

For localization with cellular markers, the BHK-21 cells were transfected with plasmid pGP2/GP3/GP4 or with GP3-HA in pMDS or with GP2-myc in pMDK. At 22 h p.t., the cells were subjected to immunofluorescence as described above with the following antibodies: rabbit polyclonal anti-HA tag antibodies (1:500, ab9110, Abcam, UK), mouse monoclonal anti-HA tag antibody (1:200, 16B12, Enzo Life Sciences, Farmingdale, NY, USA), rabbit anti-E antibodies, mouse monoclonal anti-PDI (1:250, 1D3, Enzo Life Sciences, Farmingdale, NY, USA), mouse monoclonal anti-membrin (1:100, 4HAD6, Enzo Life Sciences, Farmingdale, NY, USA), and mouse monoclonal anti-ERGIC (1:150, OTI1A8, Enzo Life Sciences, Farmingdale, NY, USA). These cells were then washed 3 times with PBS and incubated with secondary antibodies (1:800, goat anti-mouse IgG H&L Alexa Fluor 568 and 1:800 goat anti-rabbit IgG H&L Alexa Fluor 488, Abcam, UK). After immunostaining in all the cases, the cells were washed three times with PBS. Finally, the stained cultures were mounted on glass slides in a fluoroshield mounting medium also containing DAPI (Abcam, Cambridge, UK) and stored at 4 °C.

Each co-localization experiment was conducted at least 2 times, and at least 10 cells per experiment were taken to quantify the co-localization in 3D in Huygens Professional version 20.10.1p2 (Scientific Volume Imaging, Hilversum, The Netherlands). Manders’ overlap coefficient and Pearson’s correlation coefficient graphs were generated with Prism software (GraphPad Software, San Diego, CA, USA).

## 3. Results

### 3.1. GP4 Cannot Be Expressed with FLAG Tag but Is Expressed with V5 Tag Separated with a Linker Sequence. Each EAV Spike Component Is Expressed under CAG Promoter

The MultiMam system was used to generate multi-cassette plasmid containing all three genes of the arterivirus spike. Each gene with a different tag sequence was cloned into acceptor and two donor vectors designed in the system. Next, the vector plasmids were fused with cre-recombination and the obtained clones were selected using antibiotic resistance. Finally, the recombined multi-cassette plasmid contained all three genes controlled by a different promoter (Figure 1). The multi-cassette recombined plasmid pGP2/GP3/GP4 was transiently expressed in mammalian cells.

Initially, we attached a FLAG-tag to the C-terminus of GP4 because it is commonly used for the detection and purification of proteins in mammalian expression systems. The GP4 gene was cloned into an acceptor vector with a CAG promoter. However, the expression of GP4-FLAG was not detectable by Western blot in transiently transfected BHK-21 cells (Appendix A). We hypothesized that the FLAG tag fused directly to the C-terminus of GP4 might influence protein folding and therefore a linker sequence was added between GP4 and the FLAG tag. The same linker sequence allowed the expression of GP4-YFP protein previously [11]. However, the GP4-linker-FLAG construct was also not expressed in BHK-21 cells (Appendix A). Therefore, a new construct was designed, composed of GP4 with the same linker sequence but fused to the V5 tag. The sequence was cloned into pACEMam2. The GP4-linker-V5 construct was expressed in BHK-21 cells (Figure 2A).

The other viral proteins were first cloned into donor vectors: GP2-myc into pMDC (with CMV promotor) and pMDK (with CAG promotor) and the E protein gene into pML-DAZ2 (with CMV promotor) and into pMDK. GP2-myc and the E protein were expressed in BHK-21 cells only from the vector with the CAG promotor but not from the pMDC vector that had a CMV promotor (Figure 2B). The GP3-HA construct was tested only for the expression from vectors with the CAG promotor, and the double band of heterologously N-glycosylated protein was detected (Figure 2C). For the construction of the multi cassette plasmid, the cre recombination was only performed with donor and acceptor vectors with the CAG promotor as schematically shown in Figure 1. For this reason, the E construct was excluded from multicassete vector, as in the multi mam system, there are only three vectors with the CAG promotor (Figure 1).

Transient transfection of BHK-21 cells with the cre-recombined vector pGP2/GP3/GP4 revealed that all three EAV spike proteins were expressed and had the expected molecular weights (Figure 2). Only in the case of GP2-myc was the expression level in the presence of the other proteins significantly higher compared to GP2-myc expressed alone (Figure 2B). Because the E protein is probably functionally linked with the GP2/GP3/GP4 spike, co-expression of the trimer with E protein was also performed. The expression of the spike components was not positively affected by the presence of the E protein, and the expression of GP2-myc and GP4-V5 was even lower, probably because less pGP2/GP3/GP4 plasmid DNA was used for transfection (see the experimental settings). Likewise, the expression levels of E did not change upon coexpression with the trimer.

### 3.2. No Effect of Co-Expression of GP2, GP3, and GP4 on N-Glycosylation

Next, we tested GP2, GP3, and GP4 for N-glycosylation and whether the glycosylation pattern was affected by the co-expression of all proteins. We transfected BHK cells with plasmid containing just one of each gene encoding minor EAV protein or with the pGP2/GP3/GP4 plasmid. Deglycosylation was performed with Endoglycosidase H, which only cleaves mannose-rich carbohydrates, and PNGase F (Peptide-N-Glycosidase F), which cleaves all types of N-linked carbohydrates. The molecular weight of all three proteins was reduced upon incubation with Endo-H and PNGase-F proportionately to the number of their oligosaccharide side chains. This indicates that all three proteins are N-glycosylated, mainly with mannose-rich carbohydrates. The MW of Gp2-myc was reduced from 24 to 20 kDa after deglycosylation with both enzymes, consistent with the presence of one mannose-rich carbohydrate (Figure 3A). No difference in the glycosylation pattern was seen when GP2-myc was expressed together with GP3-HA and GP4-V5. GP3-HA appears in untreated samples as the characteristic double band, with an MW of 35 kDa, which is due to substochiometric N-glycosylation at the overlapping sequon NNTT close to the signal peptide [11]. In accordance, upon treatment with PNGase F, only 1 band with a MW of approximately 15 kDa was visible. After cleavage with Endo-H, two bands appeared. The minor band has the same MW as GP3 deglycosylated with PNGase-F, and the major band has a higher MW, indicating that 1 of the 6 carbohydrates attached to GP3 are of the complex type (Figure 3B). The MW of GP4-V5 was reduced from 23 to 16 kDA upon deglycosylation with both enzymes regardless of whether the protein was expressed alone or from the pGP2/GP3/GP4 plasmid. The small difference in the SDS-PAGE mobility might be due to differences in the site within the carbohydrate side chain where both enzymes cleave. While PNGase F removes the entire N-linked glycans, endo H leaves the first monosaccharide that tethers each oligosaccharide to the Asn residue of the polypeptide chain (Figure 3C).

We conclude that co-expression of GP2 and GP4 from the pGP2/GP3/GP4 plasmid has no effect on the processing of its carbohydrates. All carbohydrates remained Endo-H sensitive, indicating that the overwhelming majority of proteins did not reach the medial-Golgi, where the acquisition of Endo-H-resistant carbohydrates occurs [20]. In contrast, a small effect was seen upon co-expression of GP3. One minor Endo-H-sensitive carbohydrate was found to now be completely Endo-H resistant.

### 3.3. In Transiently Expressing Cells, GP2 and GP4 Form Complexes with Each Other While GP3 Might Also Form Complexes with GP2 and GP4, but the Evidence Is Less Strong

To check if the spike proteins of EAV form complexes with each other upon simultaneous expression, co-immunoprecipitation (IP) experiments were performed. Five wells of a six-well plate were transfected with pGP2/GP3/GP4 or mock transfected. For the IP experiment, one-sixth of the transfected cells were lysed with reducing buffer and the rest of cells were divided into five aliquots: three aliquots for IP with just one anti-tag antibody, and two aliquots for the control without antibodies, where only Sepharose A or Sepharose G was added.

An optimal Co-IP experiment for membrane-associated proteins may require detergent, which allows the hydrophobic regions of the transmembrane portions of the proteins to be shielded from the solvent to prevent misfolding, and the retention of complex formation with other proteins. Therefore, the Co-IP experiment was conducted with two different lysis buffers: one “classical” buffer containing NP-40 detergent and the other containing DDM detergent, which is more suitable for membrane proteins [21,22].

Immunoprecipitates were subjected to SDS-PAGE and Western blot with anti-myc, anti-HA, and anti-V5 antibodies, respectively. GP2-myc was detectable upon IP with anti-myc, and with anti-V5 and anti-HA antibodies (Figure 4A,C). GP3-HA was detected upon IP with anti-HA (Figure 4A,C). GP4-V5 was detectable upon IP with anti-V5 but only poorly upon IP with anti-myc and anti-HA antibodies (Figure 4A,C). The results from using the two detergents are very similar. Unspecific bands visible in the anti-V5 blots may come from unspecific binding from the cell lysate, as a similar-sized band appeared in some control blots with Sepharose G (Figure 4B,D). Because the band appeared in samples from cells transfected with trimer, and in mock-transfected samples, it is likely a cellular protein binding to Sepharose G. Sepharose G was added to bind anti-V5- protein complex. The unspecific bands in the experiment with NP-40 detergent in the Co-IP lines of mock-transfected cells in blots with anti-myc and anti-HA were not visible in the control samples without antibodies. They are approximately 25 kDa in size, which could be a light chain of antibody. We conclude that GP2-myc and GP-V5 form complexes with each other, and they probably also form complex with GP3-HA, as both proteins could be detected upon Co-IP with anti-HA antibody.

### 3.4. The Components of the Trimer Are Highly Colocalized with Each Other

For localization of the trimer components within transfected cells, immunofluorescence with antibodies against particular tags of the pGP2/GP3/GP4 was performed. The fluorescence pattern in the cells is mostly reticular, as expected for membrane proteins residing in the ER, but some brighter perinuclear staining was also observed (Figure 5A). To quantify the co-localization of each of the trimer components, transfected cells were stained with all possible pairs of primary antibodies and the Manders’ overlap coefficient (MOC) and Pearson’s correlation coefficient (PCC) were calculated (Appendix A). The results show that all 3 proteins colocalize and each pair had a high MOC value: for the GP2-myc with GP3-HA pair, the MOC was 0.81 ± 0.15; for the pair GP2-myc GP4-V5, it was 0.84 ± 0.12; and for the pair GP3-HA and GP4-V5, it was 0.9 ± 0.07. (Figure 5B). PCC values were also high: for the GP2-myc with GP3-HA pair, PCC was 0.74 ± 0.18; for the pair GP2-myc GP4-V5, it was 0.76 ± 0.15; and for the pair GP3-HA and GP4-V5, PCC was 0.85 ± 0.08.

### 3.5. Components of the Trimer Partially Co-Localize with E Protein

Because the E protein is functionally associated with the GP2/GP3/GP4 spike, we tested whether they co-localize in BHK-21 cells co-transfected with pGP2/GP3/GP4 and E. Fixed cells were subjected to double immunostaining with rabbit anti-E antibodies and antibody against a particular tag of the trimer complex (Figure 6). All of the spike components co-localized with E. The MOC was 0.60 ± 0.08 for the GP2-myc and E pair, 0.51 ± 0.15 for the GP3-HA and E pair, and 0.54 ± 0.09 for the GP4-V5 and E pair. The PCC was 0.53 ± 0.07 for the GP2-myc and E pair, 0.45 ± 0.14 for the GP3-HA and E pair, and 0.46 ± 0.1 for the GP4-V5 and E pair. Likewise, the level of co-localization of the E and spike component was investigated in BHK-21 cells infected with the EAV Bucyrus virus strain and subsequently transfected with pGP2/GP3/GP4. The MOC was 0.64 ± 0.08 for the GP2-myc and E pair, 0.59 ± 0.11 for the GP3-HA and E pair, and 0.65 ± 0.11 for the GP4-V5 pair and E pair. The PCC was 0.5 ± 0.09 for the GP2-myc and E pair, 0.51 ± 0.13 for the GP3-HA and E pair, and 0.54 ± 0.13 for the GP4-V5 pair and E pair.

Regardless of whether cells were transfected or infected, the levels of co-localization of E with the components of the trimer were lower than the values obtained for the co-localization of trimer components with each other.

### 3.6. In Cells Transiently Expressing the EAV Spike, GP2-Myc Localizes to ER, Cis-Golgi, and ERGIC While GP3-HA Localizes Mainly in ER, to Some Extent in Cis-Golgi, and Minimally in ERGIC

In the next step, we investigated the co-localization of GP3-HA with markers specific for compartments of the exocytic pathway. BHK-21 cells were transfected with pGP2/GP3/GP4, and the cells (22 h p.t.) were fixed and subjected to double immunostaining with rabbit anti-HA antibodies and antibody directed against one of the following markers of the secretory pathway: protein disulfide-isomerase (PDI) for ER; ERGIC-53 for the ER-Golgi intermediate compartment, ERGIC; and membrin for cis-Golgi. The results show that for GP3-HA, the co-localization was the highest, with ER marker MOC = 0.71 ± 0.1 and PCC = 0.62 ± 0.12, followed by the cis-Golgi marker (MOC = 0.6 ± 0.08; PCC = 0.49 ± 0.02) (Figure 7A,B), and the lowest for the ERGIC compartment marker (MOC = 0.41 ± 0.08; PCC = 0.23 ± 0.08) (Figure 7C).

The localization of GP2-myc was analyzed in the same manner. GP2-myc co-localized with ER (MOC = 0.79 ± 0.07, PCC = 0.7 ± 0.01), and with cis-Golgi (MOC = 0.7 ± 0.1, PCC = 0.71 ± 0.09) (Figure 7A,B), and to some extent with the ERGIC compartment marker (MOC = 0.57 ± 0.12, PCC = 0.66 ± 0.12) (Figure 7C).

To check whether GP2-myc and GP3-HA remain in the ER if expressed separately, the same experiment was conducted with cells transfected with plasmid encoding just one gene. Interestingly, while the levels of co-localization with ER were similar to those in pGP2/GP3/GP4, the co-localization with cis-Golgi and ERGIC was reduced. The GP2-myc was co-localized with ER (MOC = 0.9 ± 0.03, PCC = 0.73 ± 0.09), weakly with cis-Golgi (MOC = 0.64 ± 0.09, PCC = 0.45 ± 0.01) (Figure 8A,B), and to some extent with the ERGIC compartment marker (MOC = 0.57 ± 0.08, PCC = 0.33 ± 0.05) (Figure 8C). GP3-HA was co-localized with ER (MOC = 0.85 ± 0.07, PCC = 0.67 ± 0.14), to some extent with cis-Golgi (MOC = 0.58 ± 0.06, PCC = 0.39 ± 0.1) (Figure 8A,B), and weakly with the ERGIC compartment marker (MOC = 0.52 ± 0.1, PCC = 0.33 ± 0.07) (Figure 8C).

## 4. Discussion

Recombinant viral spikes are a great tool for viral research, such as vaccine and antiviral drug development, virus neutralization assays, diagnostic testing, and other scientific purposes, such as structure determination of the protein [23,24,25,26]. Proteins can be over-expressed in bacteria, yeast, insect, and mammalian host systems, but usually, glycoproteins have to be expressed in eukaryotic, preferably mammalian, host systems, e.g., gonadotropin hormones [27]. In case of viral glycoproteins, such as hemagglutinin (HA) and neuraminidase of influenza virus, proteins expressed in mammalian cells elicit a better immune response with higher antibody affinity than proteins expressed in other systems [28,29]. The highly N-glycosylated spike (S) protein of SARS-CoV-2 has been successfully expressed and purified in mammalian systems and is extensively used for many ELISA assays, whereas cells expressing the spike are used for flow cytometry applications [30,31].

There are multiple ways to express complexes of proteins [32]. We chose a system that allows the expression of three proteins from the same vector (Figure 1) and hence every transfected cell should express all three proteins [33]. We thought that it might facilitate formation of the heterologous trimer composed of GP2/GP3/GP4. This is a unique feature of Arteriviruses, and most other enveloped viruses, such as Influenza and Coronavirus, contain homotrimers, HA and S, respectively, that are spontaneously formed in ER [34,35].

In our cloning strategy, we decided to add a tag to the C-terminus of all three proteins, since their N-termini contain a cleavable signal peptide. We first tagged GP4 with the FLAG tag, since it is most commonly used for mammalian membrane protein expression [36]. Despite the screening of different clones, the expression of GP4-FLAG was not detected. Since we assumed that the fusion of a tag directly to the very short cytoplasmic tail of GP4 might affect protein folding, we added a short linker, which, however, did not improve the expression. After the exchange of the FLAG with the V5 tag (the linker was kept), the expression was easily detectable, indicating that the FLAG tag itself negatively affected the expression or stability of the protein (Appendix A). Negative data on protein expression are rarely published, but recombinant glutamate dehydrogenase lost its enzymatic activity if FLAG was tagged at the C-terminus, in contrast to N-terminally tagged enzyme [37].

We also did not obtain the expression of E protein and GP2-myc from a CMV promoter, but sub-cloning of the same genes into a vector with thee CAG promotor resulted in good expression levels (Figure 2). One possible explanation for the lack of expression of pML-DAZ2 and pMDC vectors could be a very inefficient promoter.

CMV (from cytomegalovirus) and CAG (fusion of the CMV early enhancer, modified chicken β-actin promoter, and rabbit β-globin splice acceptor site) are both strong promotors for mammalian cells. The CAG promotor is considered the stronger one: in 1 study, the transfection efficiency of eGFP was 81.3% if expressed from a CAG promotor but only 59.1% for a CMV promoter. In case of viral protein production, the HBsAg (hepatitis B-soluble antigen) expression level was 36.1% higher with CAG versus CMV promoter [38].

GP2-myc and GP3-HA colocalized to a high degree with the ER marker PDI. The Pearson’s correlation coefficient was essentially the same for the ER marker PDI if the proteins were expressed from the pGP2/GP3/GP4 plasmid or if expressed alone.

The co-localization of the GP2-myc with the cis-Golgi maker membrin and ERGIC marker was much higher for GP2-myc expressed in the presence of other spike components compared to GP2-myc expressed alone. This would imply that upon co-expression with GP4 and GP3, GP2-myc is transported beyond the ER. This is consistent with the results shown in Figure 2B, where the expression of GP2-myc from pMDK was weaker than from the pGP2/GP3/GP4 plasmid. A possible explanation is that GP2-myc folding is better in the presence of GP4 and/or GP3, and hence less protein is being degraded in ER, and more is transported to the cis-Golgi apparatus. We postulate that GP2 of EAV is stabilized by oligomerization with other trimer components; alternatively, if the GP2 is expressed alone, it is misfolded and degraded. In contrast, co-expression of E with GP2/GP3/GP4 had no significant effect on their expression levels.

Furthermore, all three proteins remained Endo-H sensitive if expressed from the pGP2/GP3/GP4 plasmid, except for one carbohydrate side chain on Gp3-HA. Thus, the formation of a complex between GP2, GP3, and GP4 does not induce transport of the proteins to more distal regions of the Golgi and to the plasma membrane. This is consistent with initial studies on the processing of GP2, which remains Endo-H sensitive if expressed alone and becomes Endo-H resistant only when it is incorporated into virus particles [4].

The co-immunoprecipitation experiments showed that GP2-myc interacts with GP3-HA and with GP4-V5 since GP2-myc bands were detected after co-IP with anti-HA and anti-V5 antibodies. However, the amount of coprecipitated GP3-HA might have been underestimated as the glycosylated GP3 double band was not as visible in the Western blots as the deglycosylated protein, which was not the case for the two other glycoproteins (see Figure 3B).

The reverse Co-IP experiments revealed less co-precipitated protein. The amount of GP4-V5 co-precipitated with anti-HA and anti-myc antibodies was lower, suggesting that GP4-V5 is synthesized in excess over GP2-myc and thus only a few GP4-V5 molecules find a suitable binding partner (Figure 4A,C). The detergent used in the lysis and IP buffers seems to not have a major influence on the amount of co-immunoprecipitation of arteriviral minor glycoproteins. In the case of GP3-HA, the small amount of co-precipitated protein might be due to ineffective anti-HA antibodies, which showed only a weak signal in the Western blot if the antibody was also used for precipitation of GP3-HA (Figure 4A,C, middle blots). Nevertheless, the amount of co-precipitated proteins was very low, suggesting that only a small fraction of minor proteins form heterologous complexes.

The rather small fraction of dimeric or trimeric complexes might not be an artefact of the expression system. In EAV-infected cells, the number of heterologous complexes is also small compared to monomers or homodimers [6]. The small number of GP2/GP3/GP4 in the virion might therefore be due to the small number of complexes formed in the cell.

To the authors’ knowledge, this is the first description of the expression of an Artervirus heterotrimer in mammalian cells from a single plasmid. To purify trimeric complexes, the mammalian system needs to be scaled up. Samples likely contain a mixture of monomers and oligomers, but with the use of successive affinity chromatography steps, it will be possible to further enrich the relevant complexes, GP2/4 dimers, and GP2/3/4 trimers for subsequent structural and functional studies. The addition of the thiol oxidant diamide can induce the formation of disulfide bonds [6]. However, further studies are needed on the possible yields and functionality of recombinant arterivirus spike protein that can allow usage in downstream applications, such as structural studies, vaccine and drug development, and serological diagnostics.

Another important application is the use of a recombinant trimeric spike as a subunit vaccine, since it could be used in pregnant mares. Furthermore, it acts as a “marker vaccine” since antibodies against the tags can be used to distinguish vaccinated from infected GP5-positive animals. A marker vaccine is important for international trade, since some countries deny the import of seropositive horses or their semen, which limits the vaccination of animals and may lead to preventable outbreaks [39]. A similar vaccine might also be effective against PRRSV as current live attenuated vaccines are not ideal and are recombining with field strains [40,41].

## 5. Conclusions

The EAV minor glycoproteins were expressed with tags from a single multi-cassette vector in BHK-21 cells. For some of the proteins, the type of tag and promotor was crucial for the expression. Only a fraction of GP2/GP3/GP4 formed dimers or trimers, which is similar to studies performed previously in EAV-infected cells. The processing of N-linked carbohydrates (for all 3 proteins) and intracellular transport of GP3 and GP4 were independent of whether the proteins were expressed alone or together. The expression levels and intracellular transport of GP2 were positively affected, if it was expressed with other spike components.

## Figures and Tables

**Figure 1 viruses-14-00749-f001:**
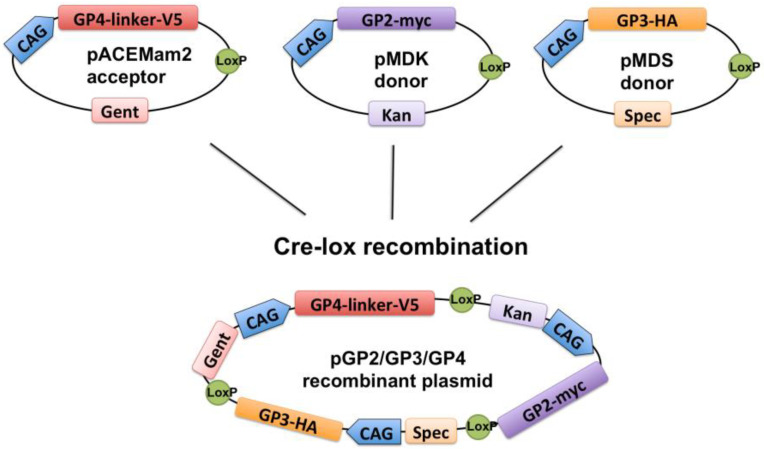
Scheme of MultiMam plasmid construction encoding the EAV spike, pGP2/GP3/GP4. Genes were cloned into one acceptor vector and two donor vectors. pACEMam2 with GP4-linker-V5 was multiplied in DH5alpha *E. coli*. Donor vectors GP2-myc in pMDK and GP3-HA in pMDS were multiplied in the pir+ *E. coli* strain. Plasmids were combined with Cre lox-recombination. Note that the depicted recombinant plasmid is just one possible combination after cre-lox recombination, and the obtained vector was not sequenced in full length to validate the order of the genes. CAG; CAG promotor, Gent: gentamycin, Kan: kanamycin, Spec: spectinomycin, LoxP: recombination sequence site.

**Figure 2 viruses-14-00749-f002:**
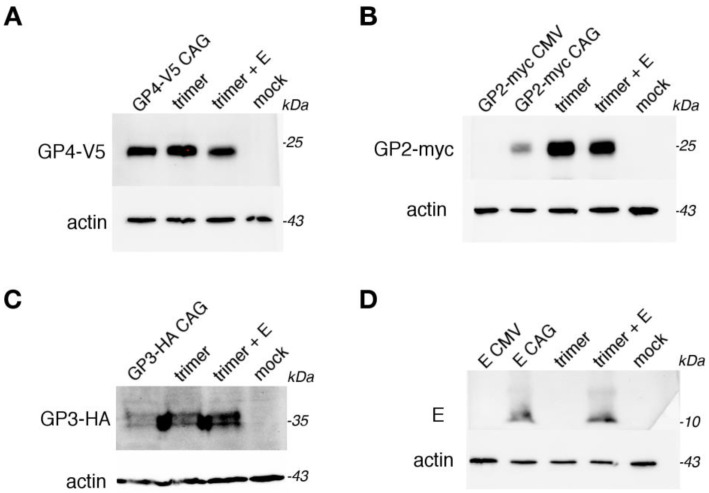
Expression of the components of the EAV spike in BHK-21 cells. BHK-21 cells were transfected with (**A**) GP4-V5 in pACEMam2; (**B**) GP2-myc in pMDC (GP2-myc CMV), GP2-myc in pMDK (GP2-myc CAG); (**C**) GP3-HA in pMDS (GP3-HA); (**D**) plasmid encoding E in pML-DAZ2 E CMV, plasmid encoding E in pMDK (CAG), or were transfected with plasmid pGP2/GP3/GP4 (trimer) or co-transfected with pGP2/GP3/GP4 and E in pMDK (trimer +E). Twenty-two hours after transfection, cells were lysed and samples subjected to SDS-PAGE and Western blotting with anti-V5 antibody (**A**), anti-myc antibody (**B**), anti-HA antibody (**C**), or anti-E antibody (**D**). Blots were cut and probed with anti-actin antibody as a loading control (**A**,**B**,**D**), or in (**C**) the blot membrane, were stripped and re-probed with anti-actin antibody. Molecular weight markers are given on the right-hand side. Mock: untransfected cells.

**Figure 3 viruses-14-00749-f003:**
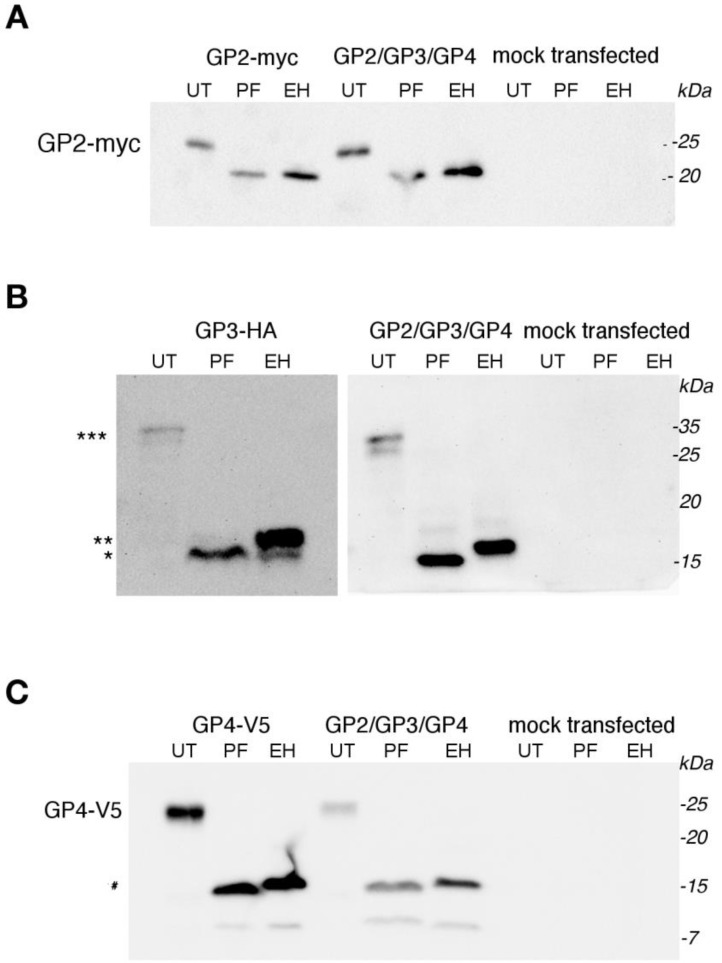
Coexpression of GP2/GP3/GP4 does not change their N-glycosylation pattern. BHK-21 cells were transfected with pGP2/GP3/GP4 (**A**–**C**) and with GP2-myc in pMDK (**A**), GP3-HA in pMDS (**B**), and GP4-V5 in pACEMam2 (**C**) or left untransfected. Then, 22 h p.t., cells were lysed in denaturing buffers, aliquots were subjected to de-glycosylation with peptide N-glycosidase F (PF), endoglycosidase H (EH), or left untreated (UT). After de-glycosylation, samples were subjected to SDS-PAGE and Western blotting with antibodies against tags of glycoproteins: anti-myc (**A**), anti-HA (**B**), and anti-V5 (**C**). *** indicates unglycosylated GP3-HA, ** not completely de-glycosylated GP3-HA, and * fully de-glycosylated GP3-HA; # indicates de-glycosylated GP4-V5. Molecular weight markers are given on the right-hand side.

**Figure 4 viruses-14-00749-f004:**
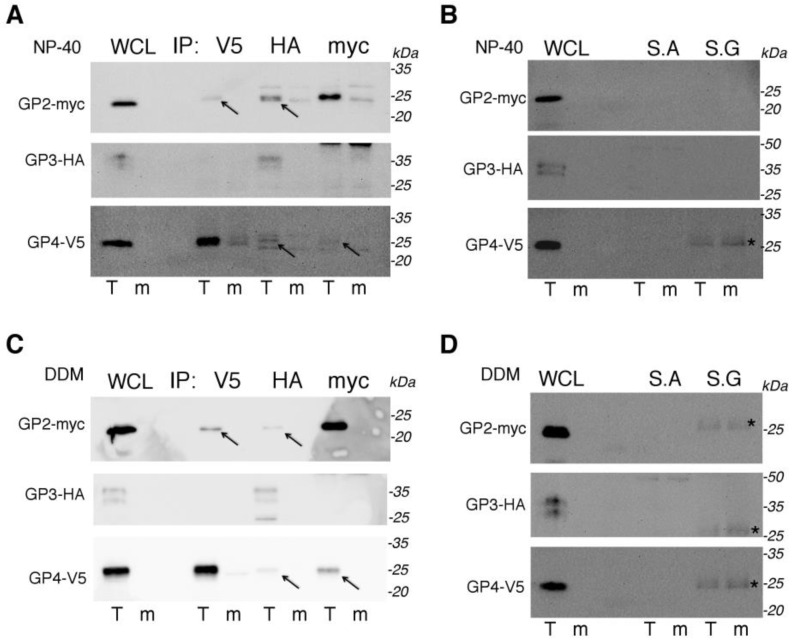
Glycoproteins of the EAV spike interact as demonstrated by co-immunoprecipitation. BHK-21 cells were transfected with pGP2/GP3/GP4 or left untransfected. Then, 22 h post transfection, cells were scraped and resuspended in ice-cold PBS. One-sixth of the cells were removed, pelleted and resuspended in SDS buffer, and boiled, as the whole cell lysate (WCL) input control. The remaining cells were lysed in two kinds of buffer. The samples depicted in (**A**,**B**) were lysed in NP-40 buffer while the samples in (**C**,**D**) were lysed in buffer containing DDM. Lysates were subjected to immunoprecipitation (IP) with mouse anti-V5, rabbit anti-myc, or rabbit anti-HA antibody (**A**,**C**). After IP, antibody-protein complexes were pulled down with either Sepharose G (anti-V5) or Sepharose A (anti-myc, anti-HA), washed, and boiled in reducing SDS buffer. (**B**,**D**) are controls of lysates incubated with Sepharose A and Sepharose G, without antibodies. Lysates and IP samples were subjected to SDS-PAGE and Western blot (WB) with anti-myc (first line pictures), anti-HA (second lines pictures), and anti-V5 antibodies (third line pictures). Arrows point to Co-IP bands. Asterisks indicate the unspecific bands that are visible in some control samples from Sepharose G lysates of trimer and mock-transfected cells. T: trimer; m: mock transfected. S.A- protein-A-sepharose, S.G-protein-G-sepharose. kDa- kilo Dalton.

**Figure 5 viruses-14-00749-f005:**
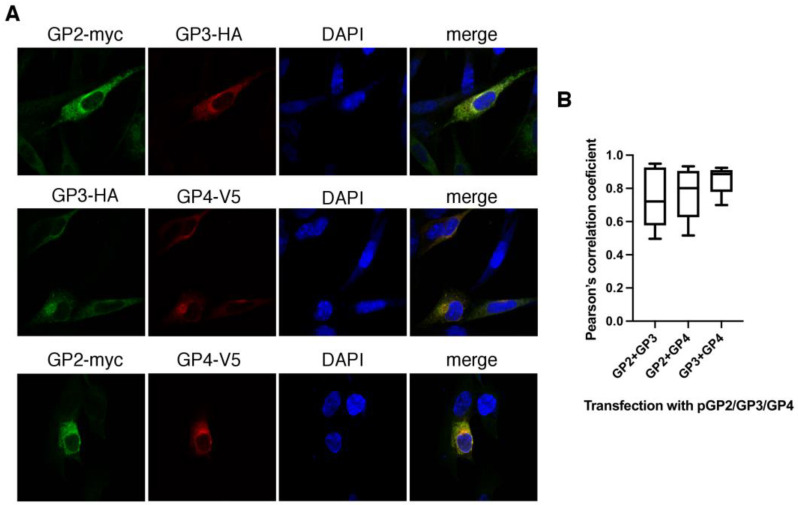
GP2-myc, GP3-HA, and GP4-V5 co-localization as shown by confocal microscopy. HK-21 cells were transfected with pGP2/GP3/GP4. (**A**) Cells were fixed 22 h post-transfection and subjected to double staining with antibody pairs against protein tags. DAPI, a nucleus stain, was added to the mounting medium. (**B**) Experiments were performed in duplicates and at least 10 cells per experiment were used to quantify the co-localization in 3D in Huygens Professional software. Pearson’s correlation coefficient graphs were generated with Prism software (GraphPad Software, San Diego, CA, USA). Untransfected cells did not show fluorescence.

**Figure 6 viruses-14-00749-f006:**
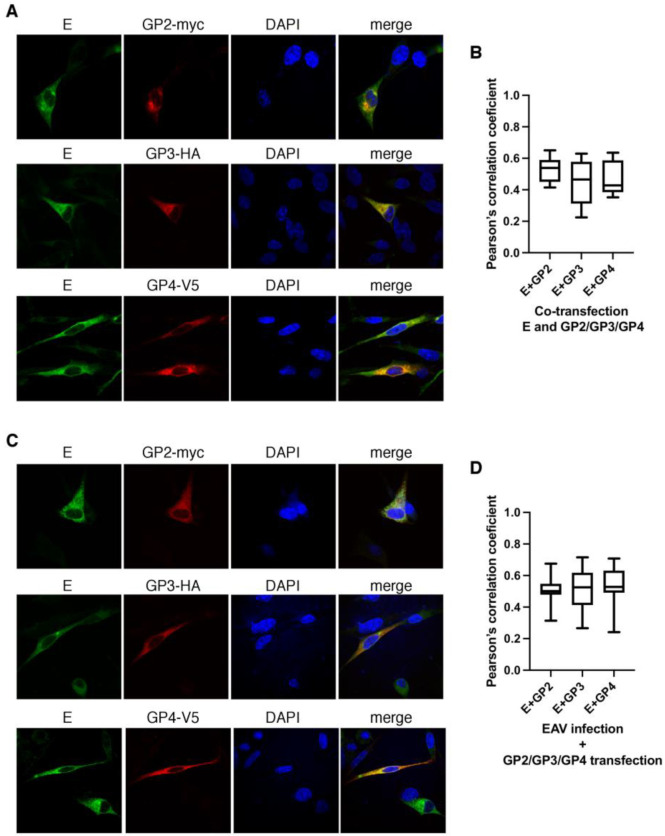
GP2-myc, GP3-HA, and GP4-V5 co-localize with E protein. BHK-21 cells were co-transfected with pGP2/GP3/GP4 and E in pMDK. (**A**,**B**) Then, 22 h after transfection, cells were fixed and subjected to immunofluorescence with rabbit anti-E (green) and mouse anti-myc (red), mouse anti-HA (red), and mouse anti-V5 (red). BHK-21 cells were first infected with the EAV Bucyrus strain at MOI 1. After 2 h, cells were washed and transfected with pGP2/GP3/GP4 (**C**,**D**). Then, 22 h after transfection, cells were fixed and subjected to immunofluorescence with rabbit anti-E (green) and mouse anti-myc (red), mouse anti-HA (red), and mouse anti-V5 (red). Experiments were performed in duplicates and at least 10 cells for each experiment were used to quantify the co-localization in 3D with Huygens Professional software. Pearson’s correlation coefficient graphs were generated with Prism software (GraphPad Software, San Diego, CA, USA). The graph shows the co-localization values for each glycoprotein with E protein in co-transfected cells (**B**), and in EAV-infected cells (**D**). DAPI: nucleus stain.

**Figure 7 viruses-14-00749-f007:**
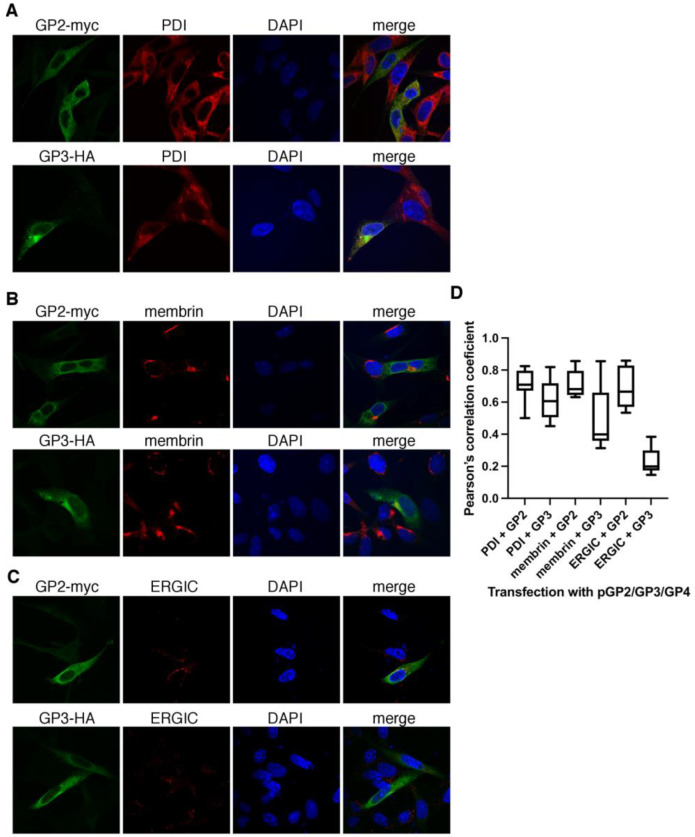
GP2-myc co-localizes to a high extent with ER, ERGIC, and cis-Golgi marker, whereas GP3-HA only co-localizes to a high extent with the ER marker. BHK-21 cells were transfected with pGP2/GP3/GP4. (**A**) Cells were fixed 22 h post-transfection and subjected to double staining with antibody against protein tags: anti-myc or anti-HA and the following organelle markers: anti-PDI for endoplasmic reticulum (ER) (**A**), anti-membrin for cis-Golgi (**B**), and anti-ERGIC53 for the ER-Golgi intermediate compartment (ERGIC) (**C**). DAPI, a nucleus stain was added to the mounting medium. The graph shows the co-localization values for GP2-myc and GP3-HA with the respective compartment markers (**D**). Experiments were performed in duplicates and at least 10 cells for each experiment were taken to quantify the co-localization in 3D in Huygens Professional software. Pearson’s correlation coefficient graphs were generated with Prism software (GraphPad Software, San Diego, CA, USA).

**Figure 8 viruses-14-00749-f008:**
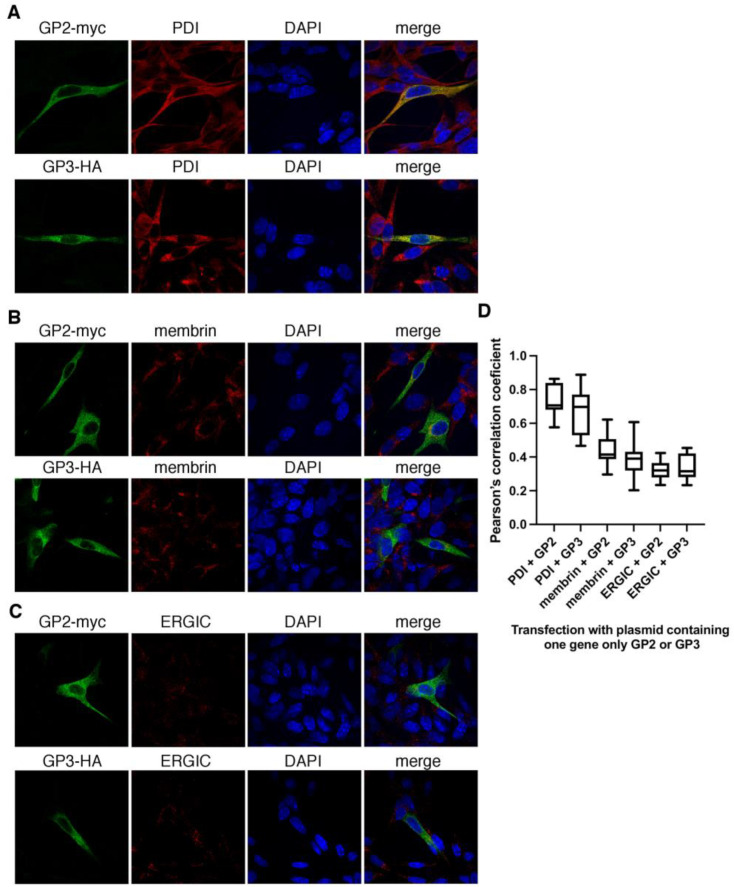
When expressed separately, GP2-myc and GP3-HA co-localize with ER, and only to some extent with cis-Golgi and weakly with the ERGIC marker. BHK-21 cells were transfected with Gp3-HA in pMDS or with GP2-myc in pMDK. (**A**) Cells were fixed 22 h post-transfection and subjected to double staining with antibody against protein tags: anti-myc or anti-HA and the following cellular markers: anti-PDI for endoplasmic reticulum (ER) (**A**), anti-membrin for cis-Golgi (**B**) and anti-ERGIC53 for ER-Golgi intermediate compartment (ERGIC) (**C**). DAPI, a nucleus stain was added to the mounting medium. Graph shows co-localization values for GP2-myc and GP3-HA with compartments marker (**D**). Experiments were performed in duplicates and at least 10 cells per experiments were taken to quantify the co-localization in 3D in Huygens Professional software. Pearson’s correlation coefficients graphs were generated with Prism software (GraphPad Software, San Diego, CA, USA).

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
