# Peer review of "Expression of the Heterotrimeric GP2/GP3/GP4 Spike of an Arterivirus in Mammalian Cells"

_viruses, 2022, doi:10.3390/v14040749_

Round 1
Reviewer 1 Report
The authors reported expression of arterivirus heterotrimer GP2/GP3/GP4 in mammalian cells to obtain a tool for viral research. The authors revised the manuscript appropriately and answered the comments by the reviewers. Unfortunately, the revised manuscript should be corrected on minor points as follows:
L179-180: The authors described protein-A-Sepharose and protein-G-Sepharose here and they used Sepharose-A and Sepharose-G in other places. Therefore the sentence (L-178-181) should be "... with protein-A-Sepharose (Sepharose-A) (Sigma-Aldrich, Poland), for ... with protein-G-Sepharose (Sepharose-G) ...".
L337-339: The authors described "... the other containing DDM dtergent, more suitable for membrane proteins". The authors should show the reference(s) on this description.
Author Response
We thank reviewer for the additional comments.
L179-180: The authors described protein-A-Sepharose and protein-G-Sepharose here and they used Sepharose-A and Sepharose-G in other places. Therefore the sentence (L-178-181) should be "... with protein-A-Sepharose (Sepharose-A) (Sigma-Aldrich, Poland), for ... with protein-G-Sepharose (Sepharose-G) ...".
Resonse:
The Sepharose A and G in brackets were added to the mansucript. Also I capitalised thw word through the manuscript as it was sometimes used with capital letter and sometimes small letter. The track changes version of manuscript is included.
L337-339: The authors described "... the other containing DDM dtergent, more suitable for membrane proteins". The authors should show the reference(s) on this description.
Response:
Reference 21 and 22 were added to the manuscript.
Reviewer 2 Report
While the authors did not perform additional experiments (both following the other's and my review) they did do some reevaluation of the data using different statistics (towards the other reviewer's comments) and added caveats to the discussion section to point out the limitations of their CoIP experiments.Author Response
While the authors did not perform additional experiments (both following the other's and my review) they did do some reevaluation of the data using different statistics (towards the other reviewer's comments) and added caveats to the discussion section to point out the limitations of their CoIP experiments.
Response:
We thank for the review, however we have to point out that the CoIP experimetns were repeated with no antibodies control. Additionally, we performed the CoIP with another suggested by the reviewer detergent: DDM. These data is presented in Figure 4. Due to Covid restrictions and many quarantine times it took few months to perform those CoIP experiments.
The evaluation of the co-localisation of the data with Pearson's coefficient was indeed suggested by another reviewer, however these changes were made already to R1 version of the manuscript.
This manuscript is a resubmission of an earlier submission. The following is a list of the peer review reports and author responses from that submission.
Round 1
Reviewer 1 Report
The authors reported expression of arterivirus heterotrimer GP2/GP3/GP4 in mammalian cells to obtain a tool for viral research. The authors successfully constructed an expression plasmid for GP2/GP3/GP4. Transient expression, heterotrimer complex formation, and colocalization in cells were reported. Experiments were scientifically designed and the results were appropriately shown. However, several failures and problems were found in the manuscript as follows:
Line105: The authors described that they used a plasmid, pDAZ from ATG:Biosynthetics. Unfortunately information about pDAZ was not found in ATG:Biosynthetics web site. What plasmid was pDAZ or pDAZ2? Cite the reference or indicate the structure of pDAZ plasmid.
Lines 111-113: Scheme shown in Figure 1 is wrong. This scheme has to be corrected. In this multiple cloning using cre-LoxP, many combinations are possible. Furthermore, the position of antibiotics genes is wrong. One of the possible plasmid structure is LoxP-Gent-CAG-PG4-linker-V5-loxP-Kan-CAG-GP2-myc-loxP-Spec-CAG-GP3-HA->the top loxP as a circular molecule. How did the authors determine the order of these genes? Did the authors confirm the sequence of the recombinant plasmid? And also the name of pACEM2 might be pACEMam2. The authors should clarify and show the correct scheme here.
Lines 124-126: It is hard to understand the sentence of “Plasmids were purified … control digested and sequenced (…) before use in experiments”. Clarify the meaning of the sentence.
Line256 and 259: The authors described that GP2-my into pMDS and … and that … not from the MDS vector that has a CMV promoter. pMDS has a CAG promoter and pMDC has a CMV promoter. Which is right, pMDC or pMDS?
Lines 270-275: The authors described the results of co-expression of the trimer with E protein. However, co-transfection of two kinds of plasmid does not always cause co-expression of two plasmid in one cell. What was the efficiency of co-expression and single-expression?
Lines 291-294: There are no panes of C and D. What was the results of blot with anti-E antibody? Legends of Figure 3 is unclear and should be corrected.
Line 338: The authors described “Each component of the EAV trimer is linked…”. According to the results, each component of the trimer is not linked by disulfide linkage. “linked” is not a suitable word.Other words should be used.
Line 385: The authors described “Components of timer …”. “timer” should be “trimer”.
Lines 391-395: The authors described the results using cells infected with EAV. How did the authors confirm the cells showing immunofluorescence were infected with EAV? Even MOI=1, not all cells were infected. The authors should show experimental evidences of the infection.
Lines403 and 406: There is no “anti-myc” but two “anti-HA”. One of “anti-HA” might be “anti-myc”.
Lines 429 and 451-452: The title of Figure 8 and 9 include GP4-V5 but there are no pictures of GP4-V5 and no description about GP4-V5 in the legends. Add pictures and explanation of GP4-V5 results or delete the word GP4-V5 from the title of Figure 8 and 9.
Lines 494-502: It is unclear what the authors discussed on GP2 and E with two promoters, CMV and CAG in this paragraph. Did the authors consider that cause of no expression of E and GP2 from CMV promoter was the efficiency of the promoter?
Lines 503-512: It is unclear what the authors discussed on differences of protein expression in three cell lines used. If the transfection efficiency was the cause of differences, the authors should show the data on transfection efficiency. And also, it is unclear why the authors mentioned on PRRSV here. Specify what the authors discuss.
Lines 513-517: This paragraph seems to be a repeat of the results and could be deleted.
Lines 554-560: It is unclear what the authors discussed on disulfide-linking between glycoproteins expressed from the plasmid. Specify what the authors discuss.
Line 559: “which then from” should be “which then form”.
In conclusion, the present manuscript will provide new insights on EAV spike proteins after correction.
Author Response
We thank the reviewer for very detailed revision of our manuscript.
Please see our responses to the comments, our responses are in green for easier read-out:
The authors reported expression of arterivirus heterotrimer GP2/GP3/GP4 in mammalian cells to obtain a tool for viral research. The authors successfully constructed an expression plasmid for GP2/GP3/GP4. Transient expression, heterotrimer complex formation, and colocalization in cells were reported. Experiments were scientifically designed and the results were appropriately shown. However, several failures and problems were found in the manuscript as follows:
Line105: The authors described that they used a plasmid, pDAZ from ATG:Biosynthetics. Unfortunately information about pDAZ was not found in ATG:Biosynthetics web site. What plasmid was pDAZ or pDAZ2? Cite the reference or indicate the structure of pDAZ plasmid.
Indeed the proper name of the plasmid is pML-DAZ2. The name was corrected throughout the manuscript. We received the plasmid from ATG:Biosynthetics, after troubles in cloning into Multi Label system. We included information that the sequence of the vector is available upon request, because we do not know if we can put it on –line. Also, a description of this plasmid is included in MM section:
“pML-DAZ2 is a 3577 bp long plasmid with CMV promotor and ampicillin resistance, its sequence is available from the authors upon request.”
Lines 111-113: Scheme shown in Figure 1 is wrong. This scheme has to be corrected. In this multiple cloning using cre-LoxP, many combinations are possible. Furthermore, the position of antibiotics genes is wrong. One of the possible plasmid structure is LoxP-Gent-CAG-PG4-linker-V5-loxP-Kan-CAG-GP2-myc-loxP-Spec-CAG-GP3-HA->the top loxP as a circular molecule. How did the authors determine the order of these genes? Did the authors confirm the sequence of the recombinant plasmid? And also the name of pACEM2 might be pACEMam2. The authors should clarify and show the correct scheme here.
In fact, we did not sequence the whole plasmid. We do not know the real structure of the clone used in this manuscript. The presence of GOIs in multi cassette vector was determined with restriction digests and PCRs. We do not think that the determination of order is important as long as all the proteins are expressed.
The Figure 1 was corrected according to the reviewers comment. We included the information that it is a schematic picture to visualise the cloning strategy, and that the order of the genes can be different.
Lines 124-126: It is hard to understand the sentence of “Plasmids were purified … control digested and sequenced (…) before use in experiments”. Clarify the meaning of the sentence.
Indeed this sentence implies that the whole plasmid was sequenced. The sentence was changed to:
„Plasmid DNA were purified (Extractme Plasmid Midi Endotoxin Free, Blirt, GdaÅ„sk, Poland), control digested to check the presence of inserts and fragments covering cloned genes were sequenced (Genomed, Warsaw, Poland) before use in experiments.”
Line256 and 259: The authors described that GP2-my into pMDS and … and that … not from the MDS vector that has a CMV promoter. pMDS has a CAG promoter and pMDC has a CMV promoter. Which is right, pMDC or pMDS?
We are sorry for the mistake. It should be that GP2-myc was cloned into pMDK (CAG) and pMDC (CMV) as indicated in section 2.2 The mistake was corrected in results section.
Lines 270-275: The authors described the results of co-expression of the trimer with E protein. However, co-transfection of two kinds of plasmid does not always cause co-expression of two plasmid in one cell. What was the efficiency of co-expression and single-expression?
Unfortunately, we did not evaluate the efficiency of co-expression and single expression in this experiment. From the immunofluorescent experiments (Figure 7A, 7B), we can conclude that only around 10% of the cells showing IF signal were expressing both E and trimer. Still we thought that in co-transfected cells there might be an effect on GP2/GP3/GP4 or the E (size of the band or expression level), but we did not see any significant result.
Lines 291-294: There are no panes of C and D. What was the results of blot with anti-E antibody? Legends of Figure 3 is unclear and should be corrected.
The legend was corrected.
Line 338: The authors described “Each component of the EAV trimer is linked…”. According to the results, each component of the trimer is not linked by disulfide linkage. “linked” is not a suitable word. Other words should be used.
We changed the sentence to:
“Each component of the GP2/GP3/GP4 form complexes with each other in transiently expressing cells, including GP3-HA.”
Line 385: The authors described “Components of timer …”. “timer” should be “trimer”.
The spelling mistake was corrected.
Lines 391-395: The authors described the results using cells infected with EAV. How did the authors confirm the cells showing immunofluorescence were infected with EAV? Even MOI=1, not all cells were infected. The authors should show experimental evidences of the infection.
The evidence for infection is expression of the E protein in cells infected with EAV. The anti-E antibody is very specific, we use it extensively and it always matches the IF signals from commercial anti-N antibody. It is true that not all the cells were infected. However, we did not want to exceed MOI 1 because cell destruction is visible at18 h p.i (strong cytopathic effect cells get rounded and detach from coverslips).
Lines403 and 406: There is no “anti-myc” but two “anti-HA”. One of “anti-HA” might be “anti-myc”.
The anti-HA was changed to anti-myc. Additionally, we spotted that the mouse anti-myc antibody from Abcam was not included in material and method section under immunofluorescent paragraph. This was also corrected.
Lines 429 and 451-452: The title of Figure 8 and 9 include GP4-V5 but there are no pictures of GP4-V5 and no description about GP4-V5 in the legends. Add pictures and explanation of GP4-V5 results or delete the word GP4-V5 from the title of Figure 8 and 9.
GP4-V5 was deleted from descriptions of the figures, as we did not perform IF with anti-V5 (we did not have anti-rabbit V5).
Lines 494-502: It is unclear what the authors discussed on GP2 and E with two promoters, CMV and CAG in this paragraph. Did the authors consider that cause of no expression of E and GP2 from CMV promoter was the efficiency of the promoter?
One possible explanation of no expression from pML-DAZ2 and pMDC vectors could be efficiency of the promoter.
We corrected the paragraph for better clarity.
Lines 503-512: It is unclear what the authors discussed on differences of protein expression in three cell lines used. If the transfection efficiency was the cause of differences, the authors should show the data on transfection efficiency. And also, it is unclear why the authors mentioned on PRRSV here. Specify what the authors discuss.
Upon quantification of transfection efficiency of YFP in different cell lines, we noticed that transfection efficiency of our BHK-21 cells is much higher than of CHO and Vero (260% higher). Therefore, the higher expression of trimer in BHK-21 cells is probably due to better transfection efficiency. Because this experiment does not contribute new information on the trimer and because in 10 days allocated for the revision of the manuscript we can not work on better transfection efficiency of other cell lines, we decided to delete this data from the revised manuscript.
Lines 513-517: This paragraph seems to be a repeat of the results and could be deleted.
Mentioned paragraph was deleted from discussion.
Lines 554-560: It is unclear what the authors discussed on disulfide-linking between glycoproteins expressed from the plasmid. Specify what the authors discuss.
This paragraph was deleted.
Line 559: “which then from” should be “which then form”.
The spelling mistake was corrected.
In conclusion, the present manuscript will provide new insights on EAV spike proteins after correction.
Reviewer 2 Report
The manuscript by Matczuk et al. shows the production of trimeric spike of Equine arteritis virus (EAV) in mammalian cells. The manuscript contains interesting data, but there are significant concerns regarding interpretation of the data.
Major concerns
- In Figure 5, the data do not support the discussed conclusions. In the western blot experiment, there is no band to support the co-IP of GP3 with GP4 (Figure 5B). The negative control shows sharp bands in figures 5B and 5C, which raise the concern about the reliability of the experiment.
- In Figures 6 through 9, the co-localization images do not appear to match the Manders’ overlap coefficient (MOC) graphs, as listed below:
- Figure 6, the co-localization images in panel A do not clearly match the data shown in the graph in panel B. The co-localization images do not show a strong co-localization of the three glycoproteins, as it was shown in the graph. The author could recalculate the degree of co-localization using Pearson correlation coefficient to confirm this data.
- Figure 7, there is appears to be little co-localization observed between E and GP4, and the MOC data does not match the result in image.
- Figure 8, there appears to be minimal co-localization of GP3 to membrin or ERGICobserved in the image. The MOC data does not match the image.
- Figure 9, the high levels of co-localization data in the MOC does not appear does not match the MOC data.
- Lines 536 -538, the authors discuss comparison of band intensities on the western blot data but do not provide quantitation to confirm the discussion points in these lines.
- Section 3.2 states “All three minor glycoproteins are expressed only in BHK-21 cells,” which suggests BHK-21 cells are the only cells in existence wherein these proteins can be expressed.
- There are many English and grammatical errors throughout the manuscript that impact its readability.
Minor concerns
- Figure 3B, the author should add molecular weight markers to the gel picture.
- Line 559, Change “from” to “form”
Author Response
We thank the reviewer for very detailed revision of our manuscript.
Please see our responses to the comments, our responses are in green for easier read-out:
The manuscript by Matczuk et al. shows the production of trimeric spike of Equine arteritis virus (EAV) in mammalian cells. The manuscript contains interesting data, but there are significant concerns regarding interpretation of the data.
Major concerns
- In Figure 5, the data do not support the discussed conclusions. In the western blot experiment, there is no band to support the co-IP of GP3 with GP4 (Figure 5B). The negative control shows sharp bands in figures 5B and 5C, which raise the concern about the reliability of the experiment.
We agree that in figure 5B the double band of GP3 with IP V5 (GP4) is very faint. There are bands in negative control in Fig. 5C IP V5 but they clearly have a different size. Potentially, this can be light chain from anti-mouse V5 antibody.
The non-uniform results of co-IP might be due to tag accessibility in each of minor protein and/or differences in antibodies binding under IP conditions.
In Figures 6 through 9, the co-localization images do not appear to match the Manders’ overlap coefficient (MOC) graphs, as listed below:
- Figure 6, the co-localization images in panel A do not clearly match the data shown in the graph in panel B. The co-localization images do not show a strong co-localization of the three glycoproteins, as it was shown in the graph. The author could recalculate the degree of co-localization using Pearson correlation coefficient to confirm this data.
We thank reviewer for this important comment. Indeed there is not complete yellow in the merge picture. This might be due to the intensity of red signal which was not so intense as green.
If there is no yellow there still might be co-localisation, which we measured here.
Picture are shown to give the view on the distribution of the protein, but quantifications are main results as it represents many cells and is unbiased.
Some pictures were changed to show more typical representations of measured cells. Changed pictures are listed below.
To make sure that the result are properly shown and interpreted we included the Pearson’s correlation coefficient data. This data was already measured together with MOC and other co-localisation parameters by Huygenes software.
Indeed, Pearson’s correlation coefficient values are slightly lower than MOCs. Especially for colocalization of GP3 with membrin and ERGIC, Pearson’s correlation coefficients (PCC) are lower. Also, there are different results on co-localisation of GP2-myc with membrin and ERGIC in cells transfected only with single gene plasmid. We comment on that in results and in discussion section.
Because PCC seems to correspond better with pictures the graphs on figures were changed to PCCs. Because many scientist prefer MOCs, the MOC graphs were kept and shown together with PCC graphs in supplementary figure 2. Also the means and SD of Pearson’s CC were included in the body of the manuscript, next to MOC data.
The reason of discrepancies in MOC and PCC co-localization measurements for ERGIC and membrin might be due to diffuse character of staining of antibodies binding to those compartments.
- Figure 7, there is appears to be little co-localization observed between E and GP4, and the MOC data does not match the result in image.
The pictures of GP4-V5 and E were changed both for co-transfection and infection to better match with quantification graph.
- Figure 8, there appears to be minimal co-localization of GP3 to membrin or ERGICobserved in the image. The MOC data does not match the image.
We think that now with the graph changed to Pearson’s correlation coefficient the pictures of GP3-HA match the data in the graph: minimal co-localisation with ERGIC and little co-localisation with membrin.
- Figure 9, the high levels of co-localization data in the MOC does not appear does not match the MOC data.
Pictures in figure 9A, GP3-HA and PDI were changed to match better with graph data.
- Lines 536 -538, the authors discuss comparison of band intensities on the western blot data but do not provide quantitation to confirm the discussion points in these lines.
We decided to delete this paragraph from discussion. It would be difficult to make valid conclusions from quantifications of band intensities from IP data, due to unknown IP and WB properties of antibodies.
- Section 3.2 states “All three minor glycoproteins are expressed only in BHK-21 cells,” which suggests BHK-21 cells are the only cells in existence wherein these proteins can be expressed.
The data on transfection of different cell lines was deleted, after suggestion from reviewer 1.
- There are many English and grammatical errors throughout the manuscript that impact its readability.
Spelling and grammatical errors were corrected through manuscript.
Minor concerns
- Figure 3B, the author should add molecular weight markers to the gel picture.
This figure was deleted from the manuscript, due to superior transfection efficiency of our BHK cells over CHO and Vero.
- Line 559, Change “from” to “form”
The spelling mistake was corrected.
Reviewer 3 Report
the authors describe the cloning and optimisation of expression of three subunits of the envelope spike protein of equine arterivirus. The manuscript is well written.
My main issue is with the CoIP experiments devised to assess the complex formation between the separate subunits.
No clear conclusions can be drawn from the represented experiments since a "no antibody" control is missing. Only very faint bands can be seen in some of the CoIP samples that could very simply be due to aspecific binding of the protein to the beads used for the pull down.
Furthermore it is unclear from the Materials section which lysis buffer exactly has been used for the CoIP experiments. The authors state that a RIPA-like buffer was used without SDS. There are quite a few different compositions described for RIPA buffers throughout literature and it is unclear which detergent was present in which concentration in the buffer used here, although one can assume NP40 or Triton X100.
An optimal CoIP experiment for membrane associated proteins requires different detergents to allow the hydrophobic regions of the transmembrane portions of the protein to be shielded from the solvent to prevent misfolding and retain complex formation with other proteins. A couple of detergents that have been found to allow for this are DDM (n-Dodecyl-B-D-maltoside) and DM (n-Decyl-β-D-Maltopyranoside). Before a confident conclusion regarding the complex formation of the subunits can be reached, the authors should perform the CoIPs using a buffer containing these detergents with the inclusion of a "no antibody" control.
Author Response
We thank the reviewer for very detailed revision of our manuscript.
Please see our responses to the comments, our responses are in green for easier read-out:
The authors describe the cloning and optimisation of expression of three subunits of the envelope spike protein of equine arterivirus. The manuscript is well written.
My main issue is with the CoIP experiments devised to assess the complex formation between the separate subunits.
No clear conclusions can be drawn from the represented experiments since a "no antibody" control is missing. Only very faint bands can be seen in some of the CoIP samples that could very simply be due to aspecific binding of the protein to the beads used for the pull down.
It is rare for the proteins to bind to the beads directly. In our lab we never used such a control.
We added the information to the discussion section that we can not exclude unspecific binding of spike proteins to the beads.
Furthermore it is unclear from the Materials section which lysis buffer exactly has been used for the CoIP experiments. The authors state that a RIPA-like buffer was used without SDS. There are quite a few different compositions described for RIPA buffers throughout literature and it is unclear which detergent was present in which concentration in the buffer used here, although one can assume NP40 or Triton X100.
Composition of the IP lysis buffer was added to the manuscript in paragraph 2.5
The composition is 25 mM Tris-HCl pH 7.4, 150 mM NaCl, 1 mM EDTA, 1% NP-40 and 5% glycerol.
An optimal CoIP experiment for membrane associated proteins requires different detergents to allow the hydrophobic regions of the transmembrane portions of the protein to be shielded from the solvent to prevent misfolding and retain complex formation with other proteins. A couple of detergents that have been found to allow for this are DDM (n-Dodecyl-B-D-maltoside) and DM (n-Decyl-β-D-Maltopyranoside). Before a confident conclusion regarding the complex formation of the subunits can be reached, the authors should perform the CoIPs using a buffer containing these detergents with the inclusion of a "no antibody" control.
We thank the reviewer for suggestion of additional experiments. We agree that the detergent might influence the IP results. Unfortunately, during 10 days we had allocated for the major revision it was impossible to purchase detergents and perform additional experiments. We think that, if there would be excess of heterologous complexes in transfected cells, we would see it even when using NP-40 detergent. With different detergent bands in co-IP may be stronger. On the other hand it might be the same. It is very strange and unique for this virus family, that minor glycoproteins are expressed in high amounts in infected cells, but only small fraction, probably even one or 2 molecules per virion are assembled (this information is taken from publications of the cro-EM pictures of the PRRSV and EAV virions). This is interesting topic and we will work on folding and oligomerisation of this proteins in the future.
We added the information in the manuscript that small amount of co-precipitated proteins might be due to detergent and that other detergent should be tried in the future.
Round 2
Reviewer 2 Report
Please more thoroughly address the first comment from the reviewer. The conclusions for Figure 5 (now Figure 4 in the revised manuscript) are questionable and the authors' rebuttal does not seem to address the reviewers' concerns.
Reviewer 3 Report
.